



# A WaveNet-Based Fully Stochastic Dynamic Stall Model

Jan-Philipp Küppers [1] and Tamara Reinicke [1]

[1]Chair of Product Development, Universität Siegen, Paul-Bonatz-Str. 9-11, 57076 Siegen, Germany

**Correspondence:** Jan-Philipp Küppers (Jan-Philipp.Küppers@Uni-Siegen.de)

**Abstract.** Accurate modeling of the dynamic stall remains a challenge for the design and construction of turbine blades and helicopter rotors. At the same time, wind turbines, for instance, are becoming steadily larger, further increasing the demands on their structure and necessitating even more detailed modeling of the forces at hand. The primarily used (semi-)empirical models today have a long research history and are invariably based on phase-averaged
data from oscillating blade pitch experiments. However, much potential for more accurate modeling of uncertainties and force peaks is wasted here, since averaging blurs many features of the response signals. Even computational fluid dynamics can help little in this regard, since the Reynolds-averaged Navier-Stokes equations used in practice cannot account for cycle variations, and scale-resolving models require extremely large amounts of computational resources. This paper presents an approach for a fully stochastic machine learning model that can nevertheless simulate these
critical properties. Aerodynamic coefficients are compared with experimental data for different test cases. It is shown that synthetic force profiles can be generated which cannot be distinguished from the experimental data visually and are very close to them in the frequency spectrum. Additionally, attention is drawn to the difficulty of evaluating such a model, as traditional error metrics are of little use. A combination of Dynamic Time Warping and the Earth Mover Distance provides a robust solution for this problem.

# 1   Introduction

As the trend towards increased performance of wind turbines continues, the calculation of the dynamic forces acting on the blades is becoming increasingly important. For classical horizontal axis turbines, a number of factors such as atmospheric turbulence, tower shadow and yaw misalignment lead to highly unsteady and nonlinear aerodynamic conditions. In vertical axis turbines, the periodic change of the angle of attack is even an inherent part of the working

principle. For all turbine types, however, there is a desire to predict loads and fatigue stresses as accurately as possible in order to build cost-effective and robust structures. Since the same phenomenon also occurs in rotary-wing aircraft such as helicopters, the common interest in these industries led to extensive research in a desire to learn more details about the mechanisms behind it ((McAlister et al., 1978), (McCroskey, 1981) and (Carr, 1988)).

The aerodynamic forces present on a wing show exceptional strong fluctuations if the flow dynamically detaches

from the suction surface. This unsteady phenomenon, called dynamic stall, is typically caused by a rapid change of the inflow conditions, such as a sudden increase in the angle of attack often in conjunction with a change in the



inflow velocity. In his very well-known publication, Carr (1988) identifies approximately 11 stages of dynamic stall. Briefly, when the static stall angle $\alpha_s$ is exceeded, flow reversal starts to occur on the surface while the boundary layer remains attached for a short amount of time and a dynamic lift overshoot occurs. The subsequent separation

process is characterized by the detachment of a dynamic stall vortex (DSV) formed near the leading edge. The vortex first remains near the leading edge above the suction surface for a short time and increases in strength until its detachment into the wake triggers the complete boundary layer detachment. The detached vortex causes a sharp decrease in pitching moment followed by a loss in lift (Müller-Vahl et al., 2017).

Modeling this phenomenon has always been a challenging task. Even the use of computational fluid dynamics

(CFD) has not produced particularly satisfactory results. Stangfeld et al. (2015) found that the Reynolds-averaged Navier-Stokes (RANS) equations, which are often used in practice, are not able to represent unsteady vortical structures. Due to their formulation, the RANS equations produce a smooth and smeared solution. Essentially, cycle-to-cycle variations are not present in the simulations, which contradicts experimental findings on pitching airfoils. They state that Largy Eddy Simulation (LES) and other scale-resolving simulation methods can be a

solution, but due to extreme computational requirements, they are often not well suited for the design of an entire wind turbine.

Another approach is to use empirical (Gormont (Gormont et al., 1973), Berg (Bianchini et al., 2016)) and semi-impirical stall models (Øye (Øye, 1990), Beddoes-Leishman (Leishman and Beddoes, 1989), ONERA (Tran and Petot, 1980)) that have been developed over the years. These models attempt to compress the entire physical process into

a set of equations that analytically return the corresponding lift, drag, and moment forces. The Beddoes-Leishman model is still considered state-of-the-art today. It models the dynamic stall effect with a set of differential equations that, divided into modules, describe different flow states, such as unsteady attached flow, unsteady separated flow and dynamic stall. Recently, there have been attempts to further improve this model by also predicting second order lift and drag forces (Bangga et al., 2020). Common to all of these models is that there is a set of static parameters

that are tuned so that the predicted results fit the phase-averaged experimental data as well as possible. This set can include up to 15 parameters, which makes it hard to tune manually.

Problematically, other researchers have found that blade pitch experiments required up to 50 cycles to converge to a mean (McAlister et al., 1978). Also, vortex shedding and recovery phases are subject to stochastic variations. Even in simple 2D cases, multiple separation flow-structures can be detected, which become even more complicated when

the patterns are viewed on a real 3D blade (Manolesos et al., 2014). Lennie et al. (2017) argue that the variations and outliers are an important part of the data set and should not be discarded by averaging. When calculating maximum aerodynamic loads, some forces could otherwise be significantly underestimated. Another argument is that at some point it becomes too difficult for a human to build a model complex enough to fit all flow regimes.

This is where the field of machine learning, which has become very popular in recent years, comes into focus.

First attempts in the past with surrogate models deal with the fitting of Kriging models (Glaz et al., 2010). Neural Networks are used by Glaz et al. (2012) and Spentzos et al. (2006) to predict unsteady RANS data. Tatar and Sabour



(2020) created a nonlinear reduced order dynamic stall model using a fuzzy inference system (FIS) and adaptive network-based FIS (ANFIS) to fit simulated RANS data as well. All of these publications have in common that their models are based on simulation data that roughly corresponds to the phase-averaged data from the measurements

discussed earlier. Here, however, the potential of true unsteady data is not yet used. We argue that since all dynamic stall models use experimental data to tune their parameters, one may as well use the raw experimental data directly. The presented model extracts all relevant features from the raw data itself and can make much more accurate predictions than the commonly used models. It can not only predict unsteady forces, but also allows to derive the range of fluctuations, maximum values, and frequencies. Our model is based on DeepMind's WaveNet architecture, a

model for generating raw audio waveforms (van den Oord et al., 2016a). Since audio data has similar 1-D time series characteristics as the wind tunnel test data, the choice was made to use the proven model in a slightly modified form. Other generative machine learning models could potentially be used as well, but are not explored in this research.

The work is organized as follows. First, Chapter 2 demonstrates the architecture and mathematical foundations of the neural network. Building on this, Chapter 3 shows how the experimental raw data is processed so that it can be

fed to the model. Chapter 4 describes the challenges of evaluating such a model and describes how the best learning parameter combination was found. Then, in Chapter 5, the dynamic stall results of the model for three different test cases are presented. The results of one test case are further clustered in Chapter 6, followed by a brief discussion of the method and an outlook in Chapter 7, ending with a conclusion in Chapter 8.

## 2   The Architecture

The Neural Network architecture used here is based on a Convolutional Neural Network called WaveNet (van den Oord et al., 2016a). It itself was inspired by PixelCNN (van den Oord et al., 2016b), a network that completes images pixel by pixel based on previously known color information. Unlike PixelCNN, which works with 2D RGB images, WaveNet processes one-dimensional audio waveforms, ergo time series. The model is generative and predicts a conditional probability distribution for sample $x_t$ based on a set of past samples $x = \{x_1, ..., x_{t-1}\}$. Thus, the

probability distribution resulting from the sequence $x$ can be derived from the conditional probabilities of each sample given its previous samples by application of the chain rule, as follows (boilard et al., 2019):

$$p(x) = \prod_{t=1}^{T} p(x_t \mid x_1, ..., x_{t-1}) \tag{1}$$

The most important features of such a model are that it is autoregressive, i.e., it generates new samples based on values that it itself has previously generated. And it is fully probabilistic, i.e. there is a probability distribution

prediction from which the final value needs to be sampled each time.

The dilated causal convolutions shown in the network architecture diagram in Figure 1 are the key idea for WaveNet to work properly. Causal here means left-side padding with zeros for all data in the network, so that the





sliding window of the convolutional filter can only access information from previous time steps and order is not violated. The dilated part describes a convolutional filter with holes that slides along the data. This enables to

drastically increase the receptive field of the network without requiring too many computationally expensive nodes. By stacking $m$ filters the distance seen into the past is doubled each time. Consequently, the dilation factors are $1, 2, 4, 8, 16, \ldots$ as shown in Figure 2. The figure also shows that for a given input, no information from the future is used to drive the calculation for the next step. Nevertheless, the model can process the entire sequence of time steps in the input vector quasi simultaneously. This vectorized processing has performance advantages during training

over serial models, such as the classic Recurrent Neural Network (RNN).

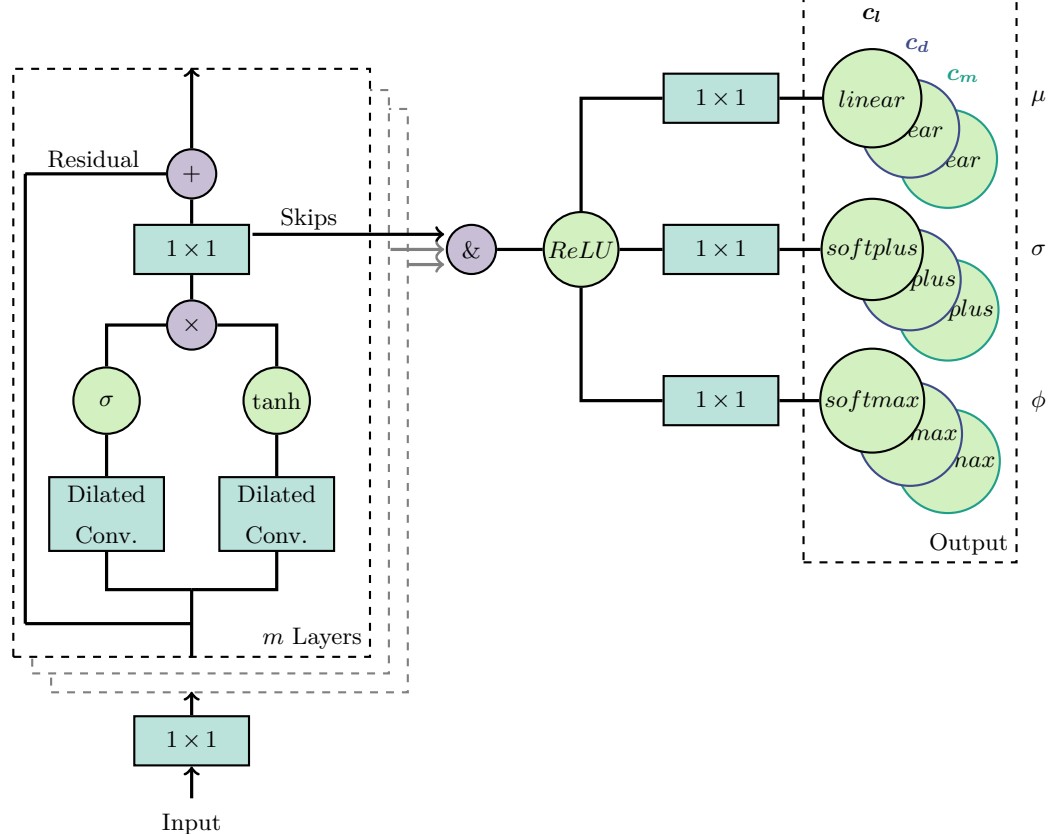

**Figure 1.** Wavenet-based model with $m$ stacked blocks. Each block containing the characteristic gated activation units combined with the skip- and residual connections. The $1 \times 1$ filters are convolutional filters with a width of 1 and are equivalent to a time-distributed, dense layer. This means, for example, that a 1 x 1 layer with 8 feature maps links the information of each time step in the same way as 8 classic fully connected nodes would do with the input of a single time step. Each coefficient needs three output nodes for the weight $\phi$ of each stacked Gaussian, the corresponding standard deviation $\sigma$, and mean $\mu$.





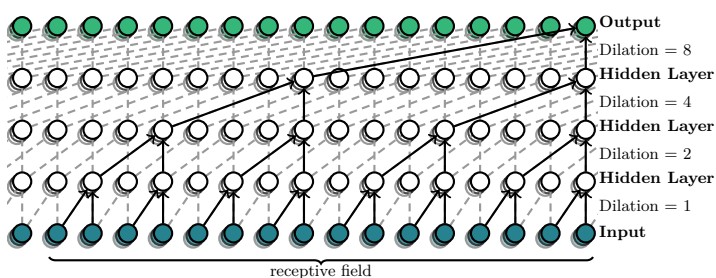

**Figure 2.** Visualization of stacked dilated causal convolutional layers. Each node in the input vector represents a single time step and the time flow is from left to right.

The original model uses quantization of the output for 8-bit integers, resulting in 256 discrete possible values. This is too imprecise for our model and introduces additional difficulties for learning performance, since the cross-entropy loss used cannot differentiate the spatial distance between the categorical buckets. Therefore, the same loss can be assigned to a close hit as to one far-off target. For our purposes it makes more sense to use a mixed density output

where we stack a user-defined amount $n$ of Gaussian $\mathcal{N}$ distributions weighted by $\phi$ which are each described by a mean value $\mu$ and standard derivation $\sigma$. This allows to approximate arbitrary conditional probability distributions (Reynolds, 2008) and is formally defined as:

$$p(y \mid x) = \sum_{i=1}^{n} \phi_i(x) \cdot \mathcal{N}(y \mid \mu_i(x), \sigma_i(x)) \tag{2}$$

where $i$ denotes the index of the corresponding mixture components. The $n$ mixture coefficients $\phi$ must sum to

one, which is reflected in the use of a softmax layer as part of the output in Figure 1. Another constraint for the Gaussian is that the standard deviation is $\sigma(x) > 0$ and is therefore using a softplus layer. The mean can take any value and is assigned to a linear layer.

The loss is described by the average negative log-likelihood of the propability density functions. First, the posterior probability is calculated by using the true solution $y$:

$$\mathcal{N}(y \mid \mu_i(x), \sigma_i(x)) = \frac{1}{\sqrt{2\pi}\sigma} \cdot exp\left[-\frac{(y-\mu)^2}{2\sigma^2}\right] \tag{3}$$

Then all posterior probabilities are multiplied with their associated weights $\phi$ to get the likelihood. After averaging the Logarithm of each result from the whole solution vector we can submit the data to an appropriate optimizer like SGD or Adam (see Equation 4). More information on the role of the skip and residual connections or the activation functions can be found in the original source (van den Oord et al., 2016a).



$$\underset{\Theta}{\arg\min} \, f(\Theta) = \frac{1}{|\mathbb{D}|} \sum_{(x,y)\in\mathbb{D}} -\log p(y \mid x) \tag{4}$$

The hyperparameters batch size, number of feature maps, number of stacked dilation filters and number of mixed Gaussians are fine tuned by a grid search for the best score. For this, the data is split into a training and test set. The test set is ignored until the final predictions are made. The training set is further divided into five parts and a k-fold cross validation is performed (Fushiki, 2011). The k-fold cross validation is a re-sampling technique used to estimate the accuracy of the model on new data. The parameter k denotes the number of folds into which the training data is divided. Then, for each hyperparameter combination from the grid search, the training is repeated k times. Each time a different fold is left out of the training and used as a validation set. The accuracy scores of all folds are averaged afterwards. This increases the reliability of estimating the subsequent accuracy of the models given new data over a simple split into training and validation data. Ultimately, we use a batch size of 30, 64 feature maps for all convolutional filters, and $n = 7$ stacked blocks, so that we look back 128 time steps, or in this context, about one full oscillation cycle. As the optimization algorithm we use the default Adam-optimizer (Chollet et al., 2015).

## 3 Experimental Data and Preprocessing

The data in this paper was originally prepared for an extensive series of experiments conducted by Hanns Müller-Vahl as part of his doctoral thesis (Müller-Vahl, 2015). The main objective was to determine whether dynamic or static blowing from two slits in the airfoil can have a beneficial effect on dynamic stall control (Müller-Vahl et al., 2015). Thus, wind tunnel tests with a 75 kW centrifugal blower were carried out in the Technion Flow Control Lab (see Figure 3). The wind tunnel is characterized by a particularly low turbulence of $0.2\,\%$ and is able to vary the flow velocity cyclically by a controlled louver mechanism to simulate gusts. A detailed description of the facility can be found in (Greenblatt, 2016).

The wing model is made of Obumodulan® and is pitched about the quarter-chord position. It has 40 surface pressure ports located in the mid-span area that are staggered to avoid interference. Piezoresistive pressure transducers are placed inside the wing to improve transient response. The experimental study relies solely on surface pressure measurements from which the instantaneous aerodynamic coefficients can be derived by integration. This means that drag forces due to friction are not considered. However, it is assumed that at the high incidence levels encountered in the experiments, the pressure-induced forces far outweigh the viscous drag forces. Overall, this is considered acceptable in the context of this work, since we are mainly interested in the corresponding lift forces. It should be noted that in the experiments presented in this section, both blowing slots were sealed with tape ($75\,\mu m$ thickness) to reduce the effects of surface discontinuity. For more information about the measurement setup, see (Müller-Vahl et al., 2016).





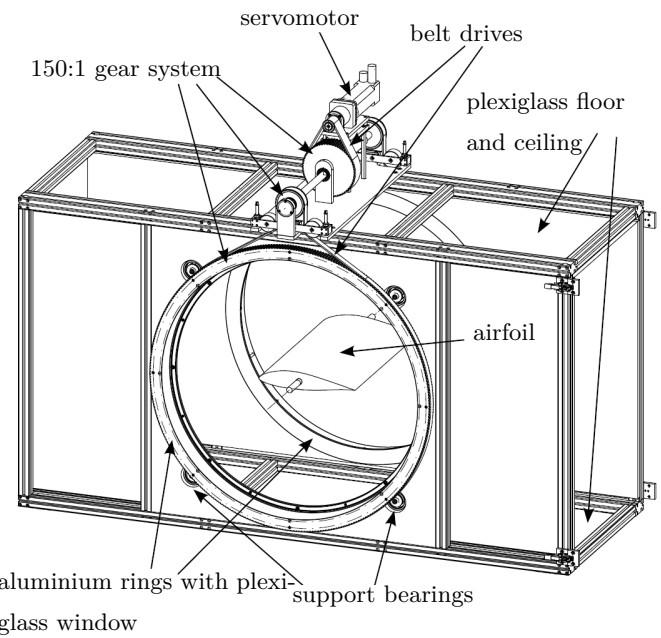

**Figure 3.** View of the test section (Müller-Vahl, 2015)

Since in this paper we want to examine standard airfoils that do not actively blow, a large part of the test data was omitted. Also, Müller-Vahl's focus was mainly on the averaged data. In the end, however, there are still 91 data-sets available for the observed S809 airfoil that meet our requirements. During the experiments, a large number of different frequencies and Reynolds numbers were recorded for a few combinations of angle of attack and amplitude. Therefore, a manual division into training and test set is necessary and makes it more difficult for the Neural Network. Otherwise the Neural Network would often practically already know the case at hand if, for example, only the Reynolds number is slightly different. An overview of the data used for training and testing can be found in Table 1.

The experimental data is sampled at a high rate of $500\,\mathrm{Hz}$ with the pitch oscillation frequency $f$, the mean angle of attack $\alpha_m$, the pitch amplitude $\alpha_a$, and the Reynolds number $Re$. The pitching motion over time $t$ can be described by the equation for the angle of attack $\alpha = \alpha_m + \alpha_a \sin(2\pi f t)$. An example of a single set can be seen in Figure 4, where the chord of $348\,\mathrm{mm}$ at $19.7\,\mathrm{m\,s^{-1}}$ flow speed resulted in a Reynolds number of 450k. To illustrate the behavior of most classic dynamic stall methods, a simulation with QBlade (Marten et al., 2013) was also added, using the Beddoes-Leishman based model implemented there. It is obvious that the model can only do what it is designed to do, which is to predict the mean lift values. Strong fluctuations in the curves are thus extremely smoothed. The model is struggling especially in the reattachment regime of the cycle, where $c_l$ is overpredicted.

Reviewing several parameter sets, it is revealed that not many interesting features are visible in the frequency spectrum beyond $30\,\mathrm{Hz}$ (see Figure 5). To reduce the computational demand and data load, the whole experimental




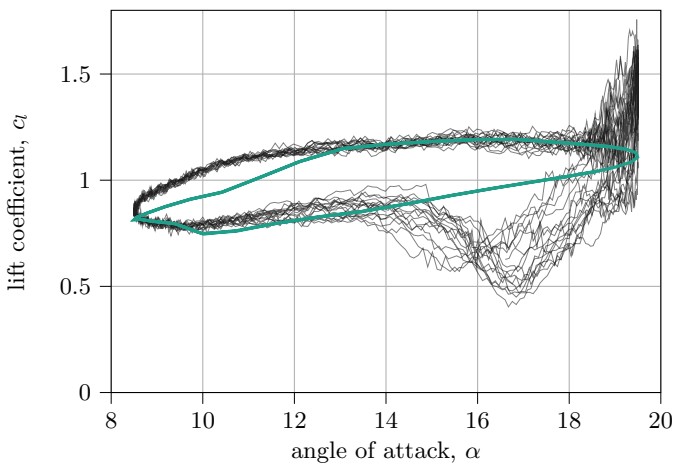

**Figure 4.** Example of raw results from the Technion Wind Tunnel tests compared to the Beddoes-Leishman Model from the QBlade package for the S809 airfoil ($f = 1.43\,\text{Hz}$, $\alpha_m = 14°$, $\alpha_a = 5.5°$ and $Re = 450000$)

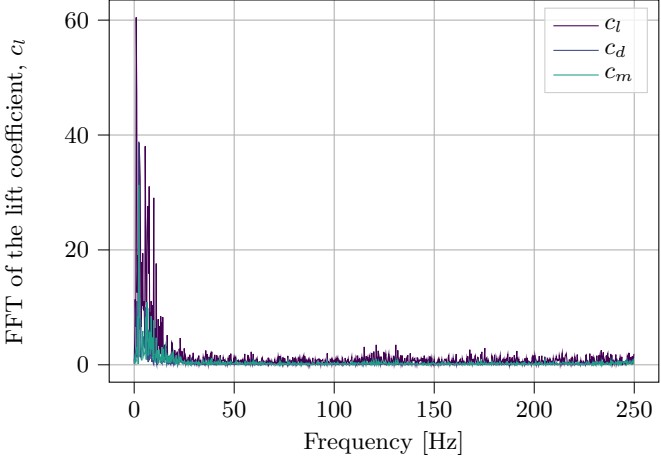

**Figure 5.** FFT of the S809 airfoil coefficients signals subtracted by their mean for $f = 1.43\,\text{Hz}$, $\alpha_m = 14°$, $\alpha_a = 5.5°$ and $Re = 450000$



**Table 1.** Available experimental data sets for the S809 airfoil sorted by the mean angle of attack $\alpha_m$. Each case consists of 3-4 repetitions with each around 120 cycles. The highlighted cases are the ones with active surge and experience a drastic flow velocity oscillation that is phase shifted by the angle $\tau$ relative to the main pitch sinusoidal oscillation.

| Cases | $\alpha_m[deg]$ | $\alpha_a[deg]$ | $f[Hz]$ | $\tau[deg]$ | $Re[-]$ |
|---|---|---|---|---|---|
| Training | 0 | 25 | 0.81 | 90, 270 | 300k $\pm$ 50% |
| | 9 | 4 | 0.67 | 0 | 300k $\pm$ 50% |
| | 12.5 | 12.5 | 0.6, 1.18 | $0, 45, \ldots, 315$ | 300k $\pm$ 50% |
| | 13.5 | 6 | 0.93 | 180 | 450k $\pm$ 21% |
| | 16.5 | 9 | 1.33 | 180 | 450k $\pm$ 24% |
| | 8 | 5.5 | 0.48, 0.94, 1.43 | | 300k, 400k |
| | 9 | 4 | 0.67 | | 330k,390k,...,570k |
| | 12.5 | 12.5 | 0.6, 0.93 | | 150k,188k,...,450k |
| | 13.5 | 6 | 0.93 | | 330k,390k,...,570k |
| | 14 | 5.5 | 0.48, 0.94 | | 390k, 450k |
| | 16.5 | 9 | 1.33 | | 330k,390k,...,570k |
| | 18 | 7 | 0.73, 1.10 | | 250k,300k,...,450k |
| | 20 | 5.5 | 0.48, 0.94, 1.43 | | 300k, 400k |
| | 21.25 | 8.25 | 1.33 | | 330k,390k,...,570k |
| Test | 10 | 10 | 1.2 | $0, 45, \ldots, 315, 57$ | 300k $\pm$ 50% |
| | 10 | 10 | 1.2 | | 150k,188k,...,450k |
| | 17 | 6 | 0.93 | | 330k,390k,...,570k |

data set is therefore downsampled by a constant factor of five to 100 Hz. The downsampling is done by a low-pass
filter and subsequent discarding of superfluous values. The highest frequency that can be represented by a Fourier
analysis afterwards is 50 Hz. It should be noted that the time vector is not a feature used in the training data.
Therefore, the model can only work with a constant time step of 0.01 s.

Each sample used as input is a small slice of specific length from the various experimental files. The length of
each slice is kept relatively short with 512 time steps, which allows a good mixing of the samples when randomly
assigning them into batches. The loss is calculated only over the part of the solution vector that has a history longer
than the receptive field (see Figure 2). To ensure that each data point is still mapped at least once with a complete
time history, the samples overlap by half. The contents of each input slice $X$ are the current angle of attack $\alpha$, the
Reynolds number $Re$ and the coefficient $c_l$ shifted by one time step (see Equation 5 and Equation 6). Therefore,
the "memory" of the model contains only the lift values, which means that the drag and moment coefficient must
be derived from the lift for the prediction. It has been found that this leads to a more reliable prediction, since the





coefficients are strongly correlated. If the model is asked to predict all coefficients based on previously self-generated data, the new samples can easily show a previously unseen pattern, making the response very noisy and degrading the training. Consequently, only the solution vector $Y$ contains the value of all coefficients for each time step. The use of other precomputed features, such as the derivative of the angle of attack, did not seem to affect the result

significantly. Thus, the model can extract the important information on its own and does not need any further guidance.

$$X = \begin{bmatrix} \alpha_2 & Re_2 & c_{l,1} \\ \alpha_3 & Re_3 & c_{l,2} \\ \vdots & \vdots & \vdots \\ \alpha_t & Re_t & c_{l,t-1} \end{bmatrix} \tag{5}$$

$$Y = \begin{bmatrix} c_{d,2} & c_{l,2} & c_{m,2} \\ c_{d,3} & c_{l,3} & c_{m,3} \\ \vdots & \vdots & \vdots \\ c_{d,t} & c_{l,t} & c_{m,t} \end{bmatrix} \tag{6}$$

The final loss is calculated by applying Equation 4 separately to each coefficient and averaging the results.

## 4   Accuracy Metric

The accuracy metric is more challenging than in the usual case for Neural Networks as we cannot use scores like the mean squared error, classification accuracy or the coefficient of determination, $R^2$. Neither a single predicted value or even a full cycle provides enough information to make statements about the quality of the model. Therefore, we need to compare the global distribution of predictions with the experimental one for each case. Thus, 60 cycles each

were predicted into the future, which corresponds to several thousand time steps. It turned out, however, that the comparison of the distributions involves further difficulties. If the distribution is put into a simple 2D-histogram and compared directly (compare Figure 8b), e.g. using the total variation (sum of absolute difference between the buckets), even a slight miss match can mean a bad score.

The Earth Movers Distance (EMD (Rubner et al., 2004) (Flamary et al., 2021), or Wasserstein Distance) is a metric

for comparing two probability distributions. It can be thought of as two piles of dirt, represented by the weighted distributions $P = \langle (p_1, \omega_{p,1}), ..., (p_m, \omega_{p,m}) \rangle$ and $Q = \langle (q_1, \omega_{q,1}), ..., (q_m, \omega_{q,n}) \rangle$ with $m$ and $n$ number of points, which are to be transformed into each other. The flow $F$ is a matrix $F = (f_{ij}) \in \mathbf{R}^{m \times n}$ where $f_{ij}$ represents the amount of dirt at $p_i$ which is matched with $q_j$. If $\mathcal{F}$ is the set of all possible flows $F \in \mathcal{F}(p, q)$ is one specific flow for which the amount of work can be calculated.





$$WORK(F, P, Q) = \sum_{i=1}^{m} \sum_{i=1}^{n} f_{ij} d_{ij} \tag{7}$$

here $d_{ij} = d(p_i, q_j)$ is the distance between $p_i$ and $q_j$. The final EMD distance is using the flow with the minimum amount of work to match $P$ and $Q$ normalized by the weight of the lighter distribution.

$$EMD(P, Q) = \frac{min_{F=(f_{ij})\in\mathcal{F}(P,Q)} WORK(F, P, Q)}{min(\sum_{i=1}^{n} \omega_{p,i}, \sum_{i=1}^{m} \omega_{q,i})} \tag{8}$$

Applied to a 2D histrogram, this means that the weights correspond to the value of the buckets and the distance

corresponds to the spatial distance. While the metric works considerably better than the total variation, it still has some disadvantages. For example, it provides a usable metric for the global distribution similarity, but ignores the frequency and appearance of the individual time series. Using a combination of Earth Movers Distance and Dynamic Time Warping (DTW (Senin, 2008) (Meert et al., 2020)) provides a solution that we discuss in the following passage.

DTW on itself is a distance measure, which is suitable to compare one-dimensional time series with each other.

The algorithm searches for the "shortest" path from the beginning to the end of both signals over an array of the pairwise distance of all points of both signals (see Figure 6). DTW has already successfully been used to prepare the clustering of cycle variations for this type of pitching airfoil experiments (Lennie et al., 2017).

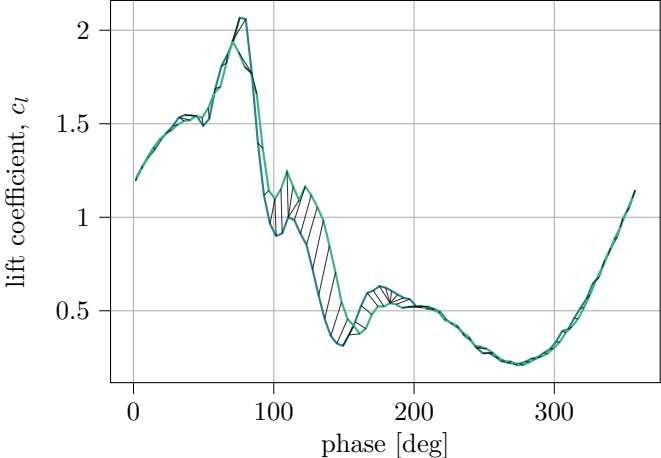

**Figure 6.** DTW distance between two representative time series. The sum of all local distances is the result.

For the full scoring, each set (predictions and experimental data) is treated as part of a bipartite graph. To prepare the EMD calculation that is essentially an optimal transport problem, the same weight $\omega$ is assigned to

each member of the graph. All weights sum up to 1. The distance $d$ between all vertices of the graph is therefore calculated by using the DTW distance (compare Figure 7) in the phase space. For this purpose, a distance matrix



is constructed that maps each time series of one set to each time series of the set to be compared. After the DTW distances between all time series are determined, it is possible to insert them into the EMD algorithm and receive a single score that accurately describes the quality of our prediction. The evaluation method, hereafter referred to as

*DTW+EMD score*, can be applied to all types of problems where distributions of time series are compared. Because of the relatively cumbersome calculation, the evaluation of the validation set is performed only every 25 epochs. If the score stagnates or increases again, the training is stopped early.

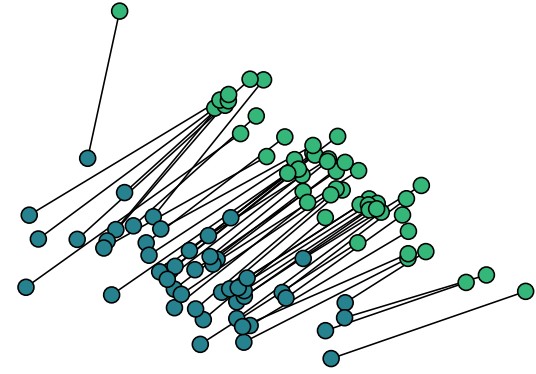

**Figure 7.** 2D visualization of optimal transport, where each point represents a time series that is part of either the predicted or experimental distribution. Instead of the Euclidean distance as shown here, the DTW distance is used in practice.

## 5 Dynamic Stall Results

The usefulness of the WaveNet-based approach is illustrated in this chapter using an oscillating pitch S809 airfoil

and comparing it to experimental data for unsteady lift, drag and moment coefficients. The pitching motion can be described by the equation $\alpha = \alpha_m + \alpha_a \sin(\omega t)$ where $\omega$ is the angular frequency corresponding to the oscillatory pitch frequency $f$. One case of surging is shown as well, here the Reynolds number is varied by the similar formula $\text{Re} = \text{Re}_m + \text{Re}_a \sin(\omega t + \tau)$. The oscillation of the surge is phase-shifted by the angle $\tau$ with respect to the normal pitch oscillation. Due to the down sampling of the experimental data, the fixed time resolution is now $\Delta t = 0.01\,\text{s}$.

In general the graphs presented are based on parameters that are not known to the Neural Network during training.

To make the different amplitudes of the coefficients comparable, all results are normalized to the range of [0,1] by min-max scaling relative to the corresponding experimental data before calculating the scores. Otherwise, it would be difficult to compare different parameter ranges, because even if their relative differences are the same, the sum of all absolute differences can still vary significantly.

Figure 8 indicates that the model accurately reconstructs the dynamic forces and is in good agreement with the higher harmonic effects. In the first case of $\alpha = 10° + 10° \cdot \sin(\omega t)$ the calculated lift curve displays a primary vortex formed by the upstroke motion that leads to the maximum lift (Point A). While the first leading edge vortex





detaches and travels downstream the lift is severely reduced (Point B). This also corresponds to a drop in pitching
moment associated with the presence of the vortex above the rear part of the upper airfoil surface (Müller-Vahl et al.,
2017)(Point C). Shortly after, a secondary vortex forms near the leading edge and increases the lift momentarily
(Point D). The subsequent break down of the vortices into smaller scale structures leads to a noisy lift response and
the lowest overall lift values (Point E). The secondary vortex is a peculiarity of the S809 profile used here and does
not occur, for example, with the NACA0018 or other profiles. The dip in lift after the secondary vortex is sometimes
slightly under predicted by the model, nevertheless as the rate of change in $\alpha$ is reduced and the incidence approaches
zero, the flow fully reattaches appropriately (Point F). Thereafter, the hysteresis curve begins again similar to the
static values with a narrow distribution during the pitch-up motion (Point G). The black arrows in Figure 8 show the
direction of time. Overall, the Neural Network reliably identifies the position of greatest uncertainty and reproduces
the range of variation. Similar patterns emerge for the coefficient of drag and momentum, where the source of the
biggest error occurs during the vortex detachment as well. The visual impression is confirmed by the low DTW +
EMD scores.

Figure 9 illustrates a similar case, but uses the special surge feature of the wind tunnel. Here the flow velocity is
strongly oscillating around $\pm 50\%$ and phase shifted relative to the main oscillation. While overall the matching of
the data indicates a decent agreement for all coefficients, some details are not correct. Particularly noticeable is
the overestimation of the lift overshoot (Point A) and the vortex shedding is triggered slightly too early (Point B).
The neural network is likely unable to gather enough information about the surge case due to the relatively sparse
training data in this parameter regime.

The third case $\alpha = 17° + 6° \cdot \sin(\omega t)$ in Figure 10 does not employ the surge feature and is in reasonably good
agreement with the measurement results. Here, the airfoil oscillates at a high angle of attack in the deep stall region.
The hysteresis curve for lift clearly shows the noisy lift overshoot during vortex detachment (Point A) and further
the point where the flow reattaches abruptly, almost stepwise, with the support of the pitch-down motion (Point B).
In addition, the distribution of the coefficients at all angles of attack is considerably broader than in the earlier cases,
which is also anticipated by the model, well seen in the heat map in Figure 10b. For the lift and drag coefficients,
a minimal offset of the values can be observed (Point C), which is likely due to inaccuracies with the non-linear
interpolation in hyperspace and can be fixed with more training samples as well. The scores reflect these difficulties
accordingly and are slightly worse than in the earlier test cases.

Another important feature that distinguishes this model from traditional methods is that the returned frequency
spectrum is close to the real spectrum as well. This opens the possibility for a more accurate analysis of blade flutter
and realistic aeroelastic responses. To illustrate this, Figure 11 shows the frequency spectrum from the first test
case next to the power spectral density estimated by Welch's method (Welch, 1967). In the frequency spectrum, the
peaks correspond to the multiples of the pitching frequency, which is imitated by the Neural Network accordingly.
The power of the simulated signal matches the experiment well. The Neural Network also recognizes the drop in the
spectral density estimate after 40 Hz , which is already at a very low power. This artifact is present in the training

**Figure 8.** Unsteady aerodynamic coefficients under dynamic stall conditions in comparison with the experimental results for the parameters $f = 1.2\,\mathrm{Hz}$, $\alpha_m = 10°$, $\alpha_a = 10°$, $Re = 300000$. The DTW + EMD score score indicates how close the two distributions are for each individual coefficient (smaller is better). 60 cycles are predicted. Figure 8b shows the global probability density function of the experiments (top) and predictions (bottom) for the unwrapped lift case in phase space.

(a) Lift DTW + EMD score = 1.44

(c) Drag DTW + EMD score = 1.17

(b) Lift *PDF*

(d) Moment DTW + EMD score = 1.46

**Figure 9.** Unsteady aerodynamic coefficients with gusts under dynamic stall conditions in comparison with the experimental results for the parameters $f = 1.20\,\mathrm{Hz}$, $\alpha_m = 10°$, $\alpha_a = 10°$, $Re = 300000 \pm 50\%$, $\tau = 90°$. The DTW + EMD score score indicates how close the two distributions are for each individual coefficient (smaller is better). 60 cycles are predicted. Figure 9b shows the global probability density function of the experiments (top) and predictions (bottom) for the unwrapped lift case in phase space.

(a) Lift DTW + EMD score = 3.94

(c) Drag DTW + EMD score = 1.56

(b) Lift $PDF$

(d) Moment DTW + EMD score = 2.05

**Figure 10.** Unsteady aerodynamic coefficients under dynamic stall conditions in comparison with the experimental results for the parameters $f = 0.93\,\mathrm{Hz}$, $\alpha_m = 17°$, $\alpha_a = 6°$, $Re = 450000$. The DTW + EMD score score indicates how close the two distributions are for each individual coefficient (smaller is better). 60 cycles are predicted. Figure 10b shows the global probability density function of the experiments (top) and predictions (bottom) for the unwrapped lift case in phase space.



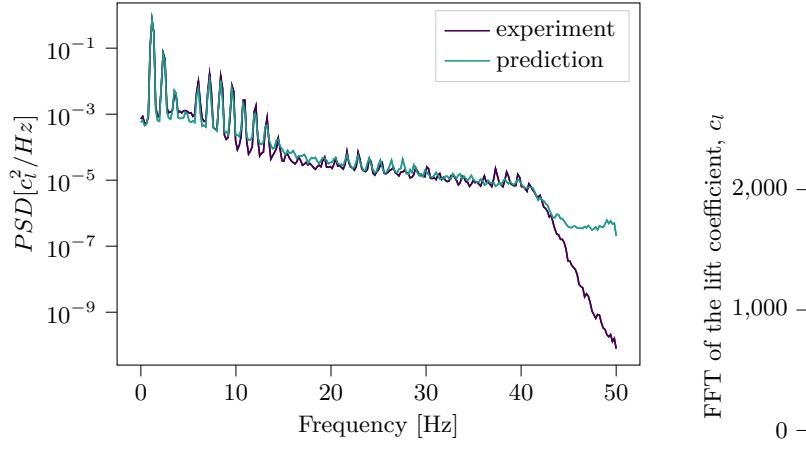

(a) Estimate of the power spectral density by Welch's method

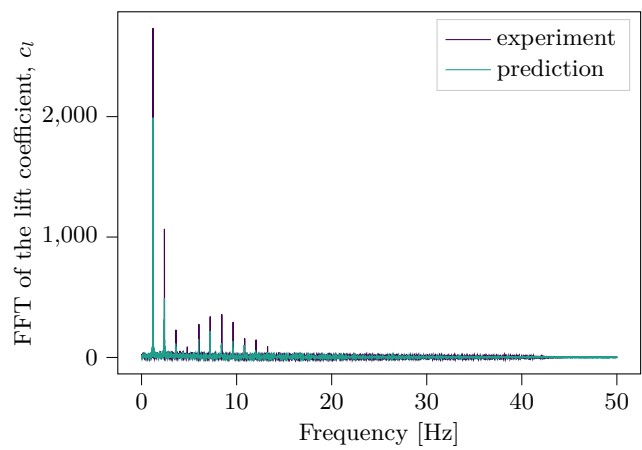

(b) Fourier transform of the lift signal

**Figure 11.** Frequency response analysis for the S809 airfoil with the parameters $f = 1.2\,\mathrm{Hz}$, $\alpha_m = 10°$, $\alpha_a = 10°$ and $Re = 300000$

data only due to down sampling, because the method used from the SciPy toolkit (Virtanen et al., 2020) employs a Chebyshev low-pass filter before removing the samples.

As far as the computational cost of applying this model is concerned, the following can be stated. The time needed to train the final model amounts to about 4 hours on an NVIDIA GEFORCE GTX 1060 6 GB. Since only one-dimensional time series are considered here, the computational time and memory consumption is certainly low compared to hardware-intensive problems, such as image recognition. Prediction requires about $0.046\,\mathrm{s}$ per time step, or about $4.6\,\mathrm{s}$ wall-clock time per second of simulated time. The prediction is thus relatively slow, since while

the model can process hundreds of parameter sets in parallel, it can only predict all sets step by step into the future. This serial mode of operation during evaluation probably has its bottleneck in the communication between CPU and GPU. However, it is still orders of magnitude faster than simulation using CFD.

## 6   On Clustering

Clustering of raw airfoil measurement data is a topic recently investigated by several authors. It is now consensus,

that the statistical mean and standard deviation used to represent cycle-to-cycle variations is inaccurate (Lennie et al., 2020). By clustering the data, group probabilities and their associated individual variances can be presented. Thus, allowing the discovery of bi- or multimodal distributions. Switching between those groups can be described as a Markov process (Ramasamy et al., 2019).





The model discussed here is able to learn and predict multimodal distributions without the need for active
switching between data-groups. However, the data available does not show obvious furcation in the coefficient data.
Nevertheless, we can cluster the time series with the method used by Lennie et al. (2020). At first we create a DTW
distance matrix between all available time series at once. Then we can apply hierarchical clustering using the Ward
method as a distance measure to form a dendrogram. The branches of the dendrogram are then cut at a specific
height that results in a user-defined number of clusters. The choice of exactly two clusters here is to some extent
arbitrary and guided only by the fact that the authors have recognized meaningful different characteristics. In this
manner physically meaningful clusters can be discovered, like the slightly different behavior after the secondary
vortex shedding in Figure 12. While one part of the time series of the experiments shows a clearly pronounced lift
overshoot, the peak is considerably weaker for the other part. When clustering the predicted data set, a similar result
is seen in Figure 13. However, whether the predicted ratios from clusters 1 and 2 are accurate is difficult to determine
with the limited amount of data. With three available measurements of 116 cycles each for this parameter set, the
ratios are spread over a wide range. The first measurement has a ratio of 62:54 for the cases with low overshoot to
those with high overshoot. The other measurements show ratios of 39:77 and 62:54, indicating a phenomenon during
the measurement and possibly worth investigating in further wind tunnel tests. The synthetic data has a ratio of
63:53.

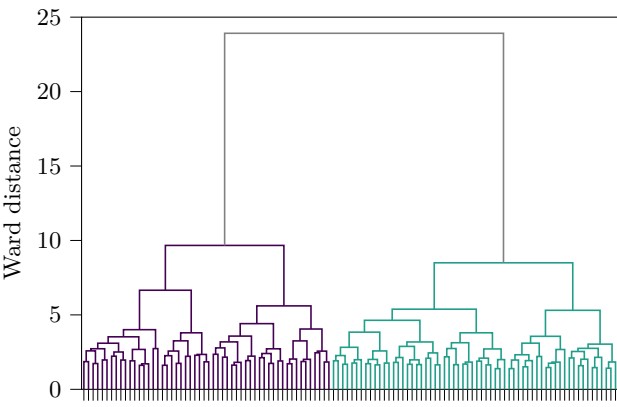

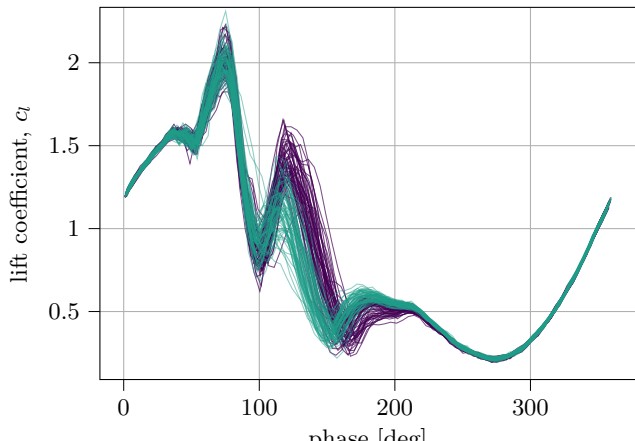

(a) Dendogram whose leaves are colored according to
the cluster membership, the length of the branches in-
dicates the distance/dissimilarity to the nearest node

(b) Cycles of the lift signal in phase space colored ac-
cording to cluster affiliation

**Figure 12.** Experimental data cluster analysis for the S809 airfoil with the parameters $f = 1.2\,\mathrm{Hz}$, $\alpha_m = 10°$, $\alpha_a = 10°$ and
$Re = 300000$. This is using the first measurement with 116 cycles resulting in a 62:54 relation for two clusters, with the first
number representing the time series with little pronounced overshoot.

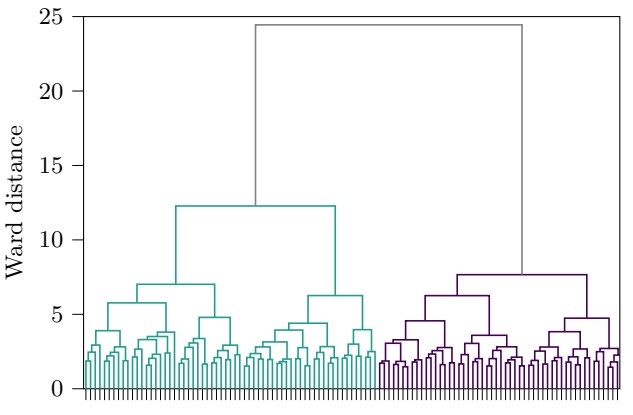

(a) Dendogram whose leaves are colored according to the cluster membership, the length of the branches indicates the distance/dissimilarity to the nearest node

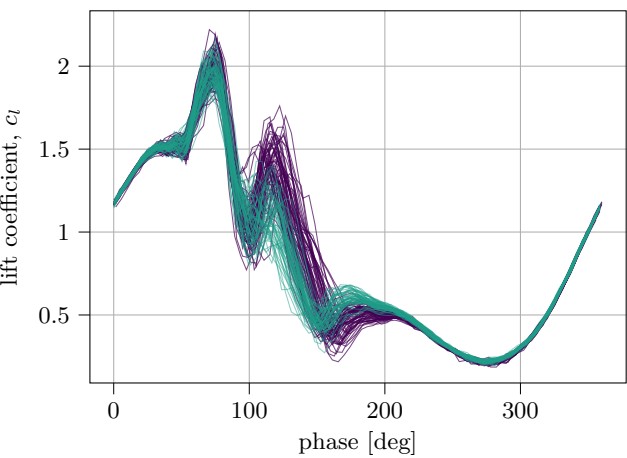

(b) Cycles of the lift signal in phase space colored according to cluster affiliation

**Figure 13.** Predicted synthetic data cluster analysis for the S809 airfoil with the parameters $f = 1.2\,\mathrm{Hz}$, $\alpha_m = 10°$, $\alpha_a = 10°$ and $Re = 300000$, the cluster ratio is 63:53, with the first number representing the time series with little pronounced overshoot.



## 7 Discussion and Outlook

While the model's capabilities are promising, its practical use is of course still limited in so far as only one blade profile can be used within a wide but still restricted parameter range. However, robust and fully functional models can be obtained by designing experiments specifically tailored to this machine learning problem. The additional information about the frequency response and possible load spectra represent a clear added value for the engineer. To be able to map a wider parameter range without gaps, more angle and oscillation frequency combinations should be used. In addition to the pitch motion, plunging could be added as a further parameter to allow the simulation of more demanding aeroelastic problems.

One has to be very sure about the quality of the training data. Since there are no subsequent plausibility checks, some major errors in the experiments would also remain in the data and predictions. Nevertheless, the model could already be incorporated into existing turbine design tools that utilize blade element theory or lifting-line theory to describe dynamic stall for an S809 airfoil.

With more data for different airfoils available global conditioning could be added to introduce geometry parameters that do not vary in time. In such a way, a very powerful and flexible model could be created to describe all types of airfoils, or even to discover novel airfoil shapes with desired characteristics through optimization. Another complex model could be created by relying on the data from the pressure ports on the airfoils surface. The derived coefficients $c_l$, $c_d$ and $c_m$ could then be calculated in a post processing step. This could potentially create a more accurate model down to the surface pressure distribution, but comes at a cost of using vastly more data and resources. Finally, the current time step could be included in the time history. In this way, a flexible change of the time step during the runtime would be possible. This could be interesting for the coupling with CFD codes or if coarser time steps are sufficient.

## 8 Conclusions

In this paper, a WaveNet-based Neural Network is established as a reduced order model for the relationship between the motion parameters of an airfoil under dynamic stall and the aerodynamic loads on it. In contrast to existing (semi-)empirical models it is fully probabilistic and working with raw wind tunnel time series. The Neural Network is autoregressive and predicts one time step at a time by generating a probability distribution from which a sample is drawn. Thus, it can predict realistic frequency responses and the local variance of the aoerodynamic coefficients. This opens up new possibilities in the study of blade flutter and other aeroelastic problems.

The presented model improves the prediction for the aerodynamic forces and their higher-harmonic effects due to vortex shedding and introduces a new level of detail, which has not been possible with traditional modeling methods. Details on the model architecture, implementation and challenges have been summarized in the present work. Three test cases were shown with different mean angles of attack, amplitude, and oscillation frequencies. The results of one case were examined in more detail for its frequency response and decomposed into clusters for comparison with



the experimental data. Finally, this work serves as a proof-of-concept for further elaboration of the method to apply stochastic machine learning models into the field of aerodynamics. The main conclusions can be summarized as 345 follows:

– Autoregressive machine learning models provide a promising base for future complex and accurate dynamic stall models.

– Fully stochastic models can present a physically realistic frequency response of the aerodynamic coefficients.

– Recovery of more raw data from old wind tunnel tests or new experiments at high sampling rates tailored to 350 machine learning are necessary to create truly flexible models.

– The phenomena detected by clustering wind tunnel data, such as furcations and bi-modal distributions of forces, can be learned by the model.

*Code availability.* The evaluation code and neural network model can be shared by contacting the corresponding author of the paper.

*Data availability.* The raw wind tunnel data and prediction results can be shared by contacting the corresponding author of the paper.

*Author contributions.* Jan-Philipp Küppers: Conceptualization, Methodology, Software, Data curation, Writing- Original draft preparation, Visualization, Investigation. Tamara Reinicke: Supervision, Project administration, Writing- Reviewing and Editing.

*Competing interests.* The authors declare that they have no known competing financial interests or personal relationships that could have appeared to influence the work reported in this paper.

*Acknowledgements.* The wind tunnel data was recorded by Hanns-Müller Vahl and David Greenblatt at the Technion - Israel Institute of Technology in cooperation with the Technical University of Berlin and was kindly provided for this project.



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
