# Peer review of "A WaveNet-Based Fully Stochastic Dynamic Stall Model"

_Wind Energy Science, 2022_

## Referee Comment (RC2)

**Review of paper wes-2022-13:**

**A WaveNet-Based Fully Stochastic Dynamic Stall Model**

by  Jan-Philipp Küppers[1] and Tamara Reinicke[1]

[1]Chair of Product Development, Universität Siegen, Paul-Bonatz-Str. 9-11, 57076 Siegen, Germany

**Brief summary**

The authors present a WaveNet-based Neural Network as a reduced order model for the relationship between the motion parameters of an airfoil under dynamic stall and the aerodynamic loads on it. In contrast to existing (semi-)empirical models typically tuned on phase-averaged data from oscillating blade pitch experiments, the new model is fully probabilistic and working with raw wind tunnel time series. The Neural Network is autoregressive and predicts one time step at a time by generating a probability distribution from which a sample is drawn.
The authors foresee that with the new model predictions of more realistic frequency responses and the local variance of the aoerodynamic coefficients it opens up new possibilities in the study of blade flutter and other aeroelastic problems.

**Overall comments**

The subject of the paper being the development of a new type of dynamic stall models based on WaveNet-based Neural Network using machine learning on raw wind tunnel data is of considerable importance for the research community. However, as also mentioned by the authors in section 7 (Discussion and outlook) its practical use, at the stage of the model presented in the paper, is still limited in so far as only one blade profile can be used within a wide but still restricted parameter range.

Therefore, a main objective with the comments from the present reviewer is to clarify what the realistic future applications of the model can be after further developments.

 **Can it replace (supplement) the use of the traditional dynamic stall models as e.g. the Beddoes Leishman model in aeroelastic simulations with codes as HAWC2, FAST or BLADED ?**

The presented prediction time of 0.046s per time step will slow down an aeroelastic simulation. **Can the simulation time be decreased by reducing the steps of the model looking backward ?**

Another limitation at the present stage seems to be that the model is locked to a time step of 0.01s.

Page 9, line 71: *It should be noted that the time vector is not a feature used in the training data. Therefore, the model can only work with a constant time step of 0.01s.*

**How difficulty is it to train in order to use different time steps ?**

The following sentence is the main argumentation from the authors of the advantages of the new model:

"*We argue that since all dynamic stall models use experimental data to tune their parameters, one may as well use the raw experimental data directly. The presented model extracts all relevant features from the raw data itself and can make much more accurate predictions than the commonly used models. It can not only predict unsteady forces, but also allows to derive the range of fluctuations, maximum values, and frequencies.*"

The use of the raw, unfiltered data has interesting potentials in modelling phenomena with stochastic response as the dynamic stall of an pitching airfoil as shown in the present paper. This also requires high quality data sets as mentioned by the authors on page 20, line 318 and could lead to a model that carries faulty data characteristics into e.g. aeroelastic simulations.

Another possibility could be to use simulated data. Its therefor surprising from the reviewers point of view that LES CFD simulations are presented as a competitive approach: page 2, line 39 *: " They state that Largy Eddy Simulation (LES) and other scale-resolving simulation methods can be a solution, but due to extreme computational requirements, they are often not well suited for the design of an entire wind turbine."*

From the reviewers point of view the use of unsteady airfoil data from high fidelity LES/DES simulations might be very interesting as input for the present model instead of/or as compliment to wind tunnel data. As an example the presented model might be able to extract post stall characteristics from 3D DES simulations[1]. In all the cases in this referenced paper the AoA was constant but the lift is fluctuating considerably due continuous vortex shedding from the separated flow. The present semi-empirical dynamic stall models are lacking the ability to generate unsteady loading for a constant AoA which might be important in aeroelastic simulations. Therefor it could be a big step forward if the described WaveNet model can do that.

**Can the authors comment on the applicability of the present model for such applications ?**

Finally as a general comment the experimental data and the derived stall characteristics presented in the paper are probably not directly applicable for aeroelastic wind turbine simulations due to: 1) the low Reynolds number and 2) the big amplitudes used in the experiments. Generally, the modern pitch regulated turbines do only accidently in operation enter into deep stall and with such big amplitudes. 1p AoA variations are typically well below 10 deg.

**Specific comments**

Line 39-41:

- **Consider to expand/modify the comments on LES simulation based on the rviewers comment above**
* * *
[1] Bertagnolio, F., Niels N. Sørensen, and Jeppe Johansen. 2006. "Profile Catalogue for Airfoil Sections Based on 3D Computations." Risø National Laboratory.

Line 130:

- Your model looks back 128 time steps which in the present case is linked to a cycle. **What guidance could otherwise be given on choosing the number of steps looking back ?**

Line 171-172:

- "*It should be noted that the time vector is not a feature used in the training data. Therefore, the model can only work with a constant time step of 0.01s.* " **Please expand on this as a fixed time step could limit the use of the model ?**

Line 271-273:

- "*Another important feature that distinguishes this model from traditional methods is that the returned frequency spectrum is close to the real spectrum as well. This opens the possibility for a more accurate analysis of blade flutter and realistic aeroelastic responses.*"

  In particular blade flutter depends strongly on the unsteady aerodynamic modelling of the linear part of the Cl curve where there is now stochastic effects. **How should the present model improve that compared e-g- with the Beddoes Leishman model ?**

**Final conclusion of review**

The reviewer can recommend publication of the paper but recommends to integrate response to the above review comments.

---

## Author Comment (AC1)

Response to the Review on the Paper wes-2022-13
**A WaveNet-Based Fully Stochastic Dynamic Stall Model**
J.P. Küppers, T. Reinicke

**We thank the reviewer for his detailed evaluation of our manuscript. Below we respond to his comments and explain the changes resulting from his comments. Please note that in the revised version of the manuscript, all changes related to reviewer #1's comments are highlighted in red if uploading a revised version is possible.**

**Comment 0:** The paper aims to provide an alternative for dynamic stall prediction to classical (semi-) empirical methods. The proposed method was constructed based on data driven approaches, adopting the DeepMind's WaveNet architecture. Overall, the paper was written well and can be followed easily. The model also produces good results with sound discussion. I enjoyed reading the whole content of the paper. Despite that, I found several issues with the paper which I would hope could be considered in the revised version of the paper.

Thank you for evaluating our work, we appreciate the effort made and your detailed insights and suggestions

**Comment 1:** Although this is minor, the usage of English needs to be checked appropriately. I found some grammatical mistakes, especially on the usage of mixed tenses.

The authors will correct these grammatical mistakes in the revised manuscript.

**Comment 2:** For a paper, the words "Chapter" does not feels right, please use "Section" instead.

Indeed, we changed the word in the revised manuscript.

**Comment 3:** Motivation to adopt a data-driven technique for dynamic stall modelling is lacking in Introduction.

We have added a paragraph highlighting the motivation of the data-driven models compared to the physically based models (l. 60ff).

**Comment 4:** Another type of simple data driven technique for optimization (such as standard gradient method, GA, etc) has been proven powerful and is practical enough to use in industry. Our group has demonstrated in (Herrmann and Bangga, J. Renew. Sustain. Ener. 2019) that this is practical enough for wind turbine design. How can we justify the real potential implementation for this approach?

The mentioned article describes an optimization of a wing profile with various optimization methods. In our case optimization only happens during the gradient descent optimization of the weights and biases of the neural network. Strictly speaking, however, the whole problem is a complex regression analysis and GA etc. is not really suitable for that. So, the rationale is that only a few methods can deliver our generative properties in the first place.

**Comment 5:** Please clearly mention the novelty of the paper.

In the introduction, we have added a paragraph that sums up the novelty of the paper and lists all the advantages.

Comment 6: How does the proposed model perform compared to a more established time series prediction models like Bi-LSTM? Or a combination of CNN-Bi-LSTM?

Based on current trends in the machine learning community, LSTMs have not received further attention. They require more memory and are considered difficult to train. It has been shown that the

autoregressive Wavenet or (more recently, after writing this paper) Transformer are even better for time series prediction[1]. Due to our limited hardware, the decision to use Wavenet was quickly made. Nevertheless, we cannot definitively assess the performance of an LSTM.

**Comment 7:** What is the size of the time series width for the selection of the window sliding method? We demonstrated in our soon to be published paper that the size of the window width plays a decisive role in the accuracy for a time series prediction. Have you made an initial study?

Yes, the size of the sliding window was part of a grid search for the best score. The vector encompassed [16, 32, 64 ,128, 256] time steps. Ultimately a receptive field of 128 steps into the past gave the best score.

**Comment 8:** The Reynolds number is fairly low for wind turbine applications. Can the model be scaled to a higher Reynolds number case?

Yes, the experiments were performed at relatively low Reynolds numbers. However, there is nothing to prevent feeding the model with further data from experiments performed at higher Reynolds numbers. If such experiments are not available, it might be possible to modify a subset of the raw data by other methods to roughly correspond to higher Reynolds numbers.

**Comment 9:** Figure 4 is not useful, please use log scale for the y axis. The magnitude of the oscillation amplitude also does not look right, a lift coefficient amplitude as large as 40 does not feel like a right value to me.

Indeed, we will use the log scale for the revised manuscript. I did not perform the experiments, but such large magnitudes can be seen in many testcases.

**Comment 10:** What is the impact of the experimental data downsampling? How if all the high frequency data is included? Will it crash due to instability? Loss in accuracy? Please justify.

Including the higher frequency components in the model would work and the model is not expected to crash. However, since above a certain threshold the main components seem to be noise, we did not expect any added value.

**Comment 11:** "Therefore, the model can only work with a constant time step of 0.01 s" I see this as a drawback. What if the user would like to choose a larger or a smaller timestep?

It would mean a slight modification of the model and a new training, but in principle it is no problem to introduce the time step as a global variable, as described in Section 7. The high sampling rate theoretically leaves a lot of room to represent significantly smaller and larger time steps in the training data. For our paper, however, this possibility was not particularly important, since it is trivial and does not add any value because the model is not intended to be used directly in practice. It would, however, imply a higher training effort.

**Comment 12:** Please check the FFT for Fig 11 (see above comment)

Fixed it in the revised manuscript as well.

**Comment 13:** As the author mentioned, the prediction is slow compared to semi empirical models, will it hinder implementation in a real wind turbine simulation tool?

While stepping forward in time is relatively "slow", the prediction can be done in parallel on large amounts of airfoil sections or even multiple turbines at once. Which should reduce the gap to the semi-empirical models considerably. The performance of the forward step can be improved by using a more powerful GPU than the consumer card from 2016 used here.

**Comment 14:** When you do the clustering using hierarchical clustering, is it possible to show the silhouette plot?

[Figure]

[Figure]

Here you go. However, I would prefer not to include them, as they lead to more clutter and even more diagrams. Mainly because I don't feel it adds much value next to the existing dendogram. If you feel it is still worthwhile, I have no problem including them in the final manuscript.

**Comment 15:** Any tests for different airfoils? Are the weights obtained here still valid?

Since we only had access to a sufficiently large data set for the S809 airfoil, the weights are unfortunately only suitable for this one. However, as we noted in Section 7, if more data were available, it would not be a problem to extend the model with airfoil-geometry related global parameters.

**Comment 16:** Last, but the most important comment, how could we adopt the model in a real wind turbine simulation tool (like Bladed, FAST, HAWC2)?

Implementing a Tensorflow model should not be too difficult. The existing programs would only need an extra interface to Python if necessary.

Then, at each time step, the recorded motion data of all airfoil sections to be simulated could be passed to the prediction function simultaneously. The turbine simulation tool then simply receives the corresponding aerodynamic coefficients and can continue to work as usual.
* * *
[1]https://bair.berkeley.edu/blog/2018/08/06/recurrent/

---

## Author Comment (AC2)

**A WaveNet-Based Fully Stochastic Dynamic Stall Model**

J.P. Küppers, T. Reinicke

**We thank the reviewer for his detailed evaluation of our manuscript. Below we respond to his comments and explain the changes resulting from his comments. Please note that in the revised version of the manuscript, all changes related to reviewer #1's comments are highlighted in purple if uploading a revised version is possible.**

**Comment 0:** Can it replace (supplement) the use of the traditional dynamic stall models as e.g. the Beddoes-Leishman model in aeroelastic simulations with codes as HAWC2, FAST or BLADED?

Potentially, this method can replace classical dynamic stall models. However, as mentioned in the paper, a more extensive database of experiments or LES simulations is required.

**Comment 1:** Can the simulation time be decreased by reducing the steps of the model looking backward ?

No, or not by much. The overhead comes from Tensorflow and the repeated call to the evaluation routine to step forward in time.

**Comment 2:** How difficulty is it to train in order to use different time steps ?

It is not very difficult and would only mean a small modification of the model. The time step could be introduced as a global variable, as described in section 7. Due to the high sampling rate, there is plenty of room to re-sample training sets with larger and smaller time steps. As already mentioned in the answer to Reviewer #1, the performance impact can be partly compensated for by the fact that the model can work very quickly in parallel. Therefore, several wing sections can be calculated simultaneously.

**Comment 3**: […] In all the cases in this referenced paper the AoA was constant but the lift is fluctuating considerably due continuous vortex shedding from the separated flow. The present semi-empirical dynamic stall models are lacking the ability to generate unsteady loading for a constant AoA which might be important in aeroelastic simulations […]. Can the authors comment on the applicability of the present model for such applications?

The model is able to predict continuous vortex shedding even at constant angles as shown in Figure 1. However, it should be noted that the training data for the model did not include static angles of attack for the airfoil. Therefore, the curves shown in Figure 1 are inferred exclusively from the dynamic data. Future experiments should therefore be sure to include static and semi-static settings to accurately model the more common range of operations for wind turbines.

[Figure]

(a) α = 10°   (b) α = 14°

Figure 1. Predicted lift coefficient for the S809 airfoil at various constant angles of attack with a short initial pitching motion (12 repetitions).

**Comment 4:** Consider to expand/modify the comments on LES simulation based on the reviewers comment

Yes, a very good idea. We added the source mentioned and suggested working with LES data when experiments are not available.

**Comment 5:** What guidance could otherwise be given on choosing the number of steps looking back ?

The decision about how many steps to look back on is appropriate should be left to a grid search. The fact that it is roughly linked to one full cycle in this case might be a coincidence. But probably a good starting point.

**Comment 6:** Please expand on this as a fixed timestep could limit the use of the model ?

As mentioned in comment 2, the model should be re-trained for real production with a variable time step. This is not a hurdle. A constant time step can still work sufficiently if it is run asynchronous to the rest of the simulation. If the flow simulation has a higher resolution in time, the aerodynamic coefficients would only be updated every 0.01s.

**Comment 7:** How should the present model improve that compared e-g- with the Beddoes Leishman model ?

As mentioned earlier. In contrast to the Beddoes Leishman model, the WaveNet model is able to simulate unsteady aerodynamic coefficients in the linear part of the CL curve.

---

## Author Comment (AC3)

Response to the Review on the Paper wes-2022-13
**A WaveNet-Based Fully Stochastic Dynamic Stall Model**
J.P. Küppers, T. Reinicke

**We thank the reviewer for his detailed evaluation of our manuscript. Below we respond to his comments and explain the changes resulting from his comments. Please note that in the revised version of the manuscript, all changes related to reviewer #1's comments are highlighted in red if uploading a revised version is possible.**

1. Please describe the basis of the window sampling size in the revised paper as well. What happens if the sampling is less than one period of dynamic stall? It would be interesting to add the initial studies in the paper. Note that in real case dynamic stall is never ever periodic especially for turbulent case, thus the model should be relatively independent of the sampling width, knowing the limit will be of importance. From what we observed in our studies, there is a certain limit of the window width needed be followed, for timeseries prediction of the turbine wake we adopted autocorrelation for finding that, but here perhaps relate that with the dynamic stall parameters as you feel more convenient.

   The data of the "initial studies" were not kept. This was an optimization process that was automated and always saved only the best models. However, even small windows sizes could give acceptable results in my experience. We found that smaller receptive fields can still provide robust solutions but maybe miss flow behavior caused by earlier events, resulting in a worse overall score. The base for the sampling size was therefore simply the best score for this particular dataset.

2. I believe the silhouette plot adds a good value in the clustering analysis.

   Perfect, we will add it to the paper.

3. Indeed we can feed the data for different airfoils and different Reynolds number. However, what about the weights obtained in the present studies? Should it be calibrated? What if we have no dynamic stall data for calibration for that airfoils? Note that in real wind turbine design load cases, manufacturers have to run more than 1000 load cases where most of the time they have no data to compare with and to re-train the models.

   If we simply do not have data for a particular wing profile, a data-driven method is obviously not to be used. Otherwise, most load cases should be within the experimental data parameter range. For extreme parameters that were not mapped in experiments, one could possibly extend the data set with LES-Simulations or artificial data.

4. No, unfortunately lift coefficient amplitude as high as 40 is totally incorrect. Just see on the raw data, the amplitude of the fluctuations is not even greater than 1. Have you checked in the FFT if you have divided the amplitude with the sampling number points in the FFT calculations?

[Figure]

Figure 1. Frequency response in normal and log representation with correct amplitude

Great observation! I actually missed to divide the amplitude by the number of sample points and was confused about the figure number you were talking about. Now it is correct.

5. The problem with a fixed time step is the real usage in wind turbine design tools. Do we need to re-train the model every single time we modify the time step? I still see this as a drawback compared to well established model like Beddoes Leishman where we do not need to bother with time step and work without any training data. Please justify this in the revised paper.

As mentioned in the first answers, it would be no problem to adapt the model to work with a variable time step. Another solution is to run the dynamics stall code asynchronously to the flow solver. One could simply take the most recent value for the lift coefficient etc. or extrapolate intermediate values for the BEM to work with.

6. As mentioned above, the problem with its generality is that the weights should be recalculated for different airfoils. For example, only for one blade we could have more than 7 different airfoils. What if we need to simulate several turbines at various inflow conditions? Coupling this with real wind turbine design tools will be a huge challenge and this should be properly mentioned in the paper.

Ideally, the whole community creates an ever growing pool of openly accessible high-resolution unsteady data (LES and experiments). Then more sophisticated models could be trained to deal with all kinds of airfoils. Another, less sophisticated solution would be to train the neural network on the difference between the experimental data and the Beddoes-Leishman model. Then the WaveNet model could be applied on top of BL for more airfoilds already. However, the question of how meaningful the data obtained in this way is still to be answered.

7. Moreover, when simulating the real turbine, we have "no initial value" to look back by 128 steps as done in the paper. I also believe this poses a challenge to use in wind turbine design tools and it is not as simple as just enabling TensorFlow as the tool. This has to be mentioned as well.

For initialization, the array was filled with artificial data. Then a short transient process takes place (see also the figures in the answers to Reviewer #2). Mostly, all parameters were simply set to "0", except for the Reynolds number.